# Attention Horizon as a Predictor for the Fuel Consumption Rate of Drivers

**DOI:** 10.3390/s22062301

**Published:** 2022-03-16

**Authors:** Hamid Sarmadi, Sławomir Nowaczyk, Rune Prytz, Miguel Simão

**Affiliations:** 1Center for Applied Intelligent Systems Research, Halmstad University, 302 50 Halmstad, Sweden; slawomir.nowaczyk@hh.se; 2Stratio Company, 1050-127 Lisbon, Portugal; rune@stratioautomotive.com (R.P.); miguelsimao@stratioautomotive.com (M.S.)

**Keywords:** attention horizon, driver performance metric, fuel consumption rate, road safety

## Abstract

Understanding the operation of complex assets such heavy-duty vehicles is essential for improving the efficiency, sustainability, and safety of future industry. Specifically, reducing energy consumption of transportation is crucially important for fleet operators, due to the impact it has on decreasing energy costs and lowering greenhouse gas emissions. Drivers have a high influence on fuel usage. However, reliably estimating driver performance is challenging. This is a key component of many eco-driving tools used to train drivers. Some key aspects of good, or efficient, drivers include being more aware of the surroundings, adapting to the road situations, and anticipating likely developments of the traffic conditions. With the development of IoT technologies and possibility of collecting high-precision and high-frequency data, even such vague concepts can be qualitatively measured, or at least approximated. In this paper, we demonstrate how the driver’s degree of attention to the road can be automatically extracted from onboard sensor data. More specifically, our main contribution is introduction of a new metric, called attention horizon (AH); it can, fully automatically and based on readily-available IoT data, capture, differentiate, and evaluate a driver’s behavior as the vehicle approaches a red traffic light. We suggest that our measure encapsulates complex concepts such as driver’s “awareness” and “carefulness” in itself. This metric is extracted from the pedal positions in a 150 m trajectory just before stopping. We demonstrate that this metric is correlated with normalized fuel consumption rate (FCR) in the long term, making it a suitable tool for ranking and evaluating drivers. For example, over weekly periods we found a negative median correlation between AH and FCR with the absolute value of 
0.156
; while using monthly data, the value was 
0.402
.

## 1. Introduction

Fuel efficiency of vehicles is an important concern for economical and environmental reasons. High efficiency of operation is the key focus of Smart Industry. It has been shown that the driver’s behavior is a significant factor in fuel efficiency of a vehicle [1,2,3,4,5,6,7]. For this reason, in the past decade there has been a surge of studies that try to determine the driving behavior that maximizes fuel efficiency [8,9,10,11,12,13]. The term “eco-driving” [8,13,14,15,16,17,18,19] is often used to describe this type of efficient driving, although it also includes strategic and tactical decisions that lead to less fuel consumption, besides driving behavior itself. In the past, relatively simple data from controlled vehicle testing was often used to determine the optimal behavior [1,20]. However, nowadays it is possible to collect sufficiently detailed data from normal vehicle operation to extract the best sequence of actions by employing fuel consumption models of vehicles, and finally optimize for the driver behavior that yields the lowest fuel consumption [21,22]. In particular, combining high-resolution sensor data with additional heterogeneous sources, such as map information, provides unique novel opportunities. The obtained driver behavior model can either be used as an algorithm for a cruise control system or for suggestions in a driver assistant system [17,18,22,23], indicating what actions to take to improve fuel efficiency. For example, there is a study that gives gear shift guidance to the driver according to the timing of the traffic lights [9]. Although cruise control and driver assistance systems can be helpful, educating the driver to correct their general driving habits has been shown to be of great effectiveness [6,15]. Eco-driving-related education is a low-cost method that can improve fuel efficiency without the need for extra equipment. Another benefit is that improving the behavior affects fuel economy even if the driver is not using the cruise control or paying attention to the assistance systems.

Analyzing the driver’s behavior to understand which decisions lead to a better fuel economy is another side of the problem. This question has been the subject of studies at least since the 1970s [2]. A key challenge in this task is to make sure that all the relevant factors are taken into account—even though capturing the full complexity of the traffic situation is usually not possible with the available data. It has also been long known [2], and even mathematically proven [10], that the specific behavior of the driver approaching a traffic light is crucial to the amount of fuel the vehicle consumes. We exploit this fact and postulate that analyzing this specific maneuver can provide insights into the driver’s more general behavior. We also demonstrate a complete solution to the problem of driver behavior ranking in such a way that it can be straightforwardly applied in realistic vehicle operation, as opposed to a controlled experimental setting. To the best of our knowledge, such a method has never been proposed before.

A different aspect of driving that could benefit from monitoring is safety. It has been shown that humans are the biggest contributing factor to all road accidents [24,25]. It is also known that variables such as trip duration and distance or number of harsh accelerations are linked to the percentage of speeding time [26], which increases the chance of crashes. After speeding, “inattentiveness” has been shown to be the biggest human error in road crashes [27]. Furthermore, reduced driver vigilance is a causative factor in 35% of vehicle accidents on motorways. Hence, it is useful to have a metric to quantify how attentive a driver is or measure his awareness, which is also the goal that we have pursued in the current study. One example of this type of work is [28], where it is proposed that the jerk (derivative of acceleration) measurement can be connected to dangerous driving behavior, and it is integrated to a mobile app that gives correction feedback to the driver.

Driver and vehicle monitoring has already been widely adopted in commercial fleets. It quantifies important aspects of vehicle operation and provides feedback loops to various actors, such as the driver or fleet operator. The general belief is that the drivers will adapt their driving style in order to obtain more favorable feedback from these systems. The fleet owner will, in return, see reduced operational costs due to lower fuel consumption and less wear and tear on the vehicles. It is inherently hard to make these systems fair, which is a significant problem. In the trivial case of comparing average fuel consumption between drivers, it is obvious that drivers who are assigned routes with higher average speed or more hilly terrain will score lower, since those two environmental conditions cause higher average fuel consumption. Therefore, it is essential to remove the impact of external conditions, such as specific route or vehicle features, from the eco-driving score when evaluating driver performance. Another possible option is to ensure that drivers are evaluated in very specific conditions in order to ensure uniformity and thus fairness.

In this paper, we introduce a new driver performance metric called *attention horizon* (AH). This metric is automatically calculated based on the onboard collected data, namely, the sequence of the acceleration and brake pedals positions before stopping at a red traffic light. The goal is to measure the awareness of the drivers with respect to the traffic conditions surrounding them. More specifically, the AH measures how far ahead drivers plan, in particular, as they start to take actions to stop the vehicle at a crossing. The key contribution of this paper is introduction of a novel descriptive driver behavior indicator, inspired by complex concepts such as “attention”, “awareness”, or “carefulness”, which is nevertheless possible to compute fully automatically, at a large scale, based on IoT data. We suggest that drivers with higher AH have more eco-friendly driving styles, which intuitively should be correlated with improved fuel efficiency. This is especially important for hybrid or electric vehicles, where harsher braking prevents optimal energy regeneration. In this study, we take advantage of naturalistic data from everyday use of heavy-duty vehicles, specifically city and coach buses, and demonstrate a meaningful correlation between the AH of a driver and their normalized fuel consumption rate (FCR) obtained during weekly and monthly periods.

The rest of this paper is organized as follows. First, our acquired data and methodology to obtain AH is described in Section 2. Then, Section 3 presents our approach to validate the correlation between AH and FCR, and describes the results. We discuss our findings and possible future work in Section 4. Finally, we conclude the paper in Section 5.

### Related Work

Recently, Carpatorea et al. have introduced an approach that analyzes the driving behavior based on the extracted sequence of drivers’ actions [29]. These actions are clusters extracted by a Gaussian mixture model from a 2D histogram (APPES) of accelerator pedal position (APP) and engine speed (ES). The clusters are represented by letter symbols, and additional symbols are defined for braking and automatic cruise control. They show that by combining their sequential feature and a support vector machine (SVM) regressor, they can predict the fuel consumption of a trip with less than 5% error for more than 60% of their samples when using the whole length of the trip.

In another study, [30], Castignani et al. take advantage of the GPS, accelerometer, magnetometer, and gravity sensors in mobile phones to detect different bad behaviors of drivers that could lead to extra emissions and fuel consumption. They train a fuzzy logic network to detect these events and use a scoring function to score the drivers based on their behavior. They show that their score is identical to drivers’ subjective scores in their controlled study. However, they fall short in linking this score to actual fuel consumption, emission, or other performance metrics.

In [31], the authors performed a literature review enumerating many different metrics that are likely related to eco-driving. They then score each type of metric related to the driver’s behavior by combining the depth of its citations and also different experts’ opinions through the grounded group decision-making (GGDM) method [32]. They remove the unimportant factors and rank other factors based on the score of their importance. However, they do not evaluate the accuracy of their ranking using any type of experiments.

Hoffman et al. [33] try to model the effect of route inclination and vehicle payload, and subtract their effect on the vehicle’s fuel consumption. Then, they take advantage of their nonlinear regression model to measure the quality of the studied truck drivers’ driving behavior using dummy variables. Although their model can rank the drivers’ behavior to some extent, it does not explain why some drivers are better than the others regarding fuel economy.

The authors of [34] define a constant and also a moving average power ratio index for drivers. The power ratio is based on the comparison between the power that is needed by the vehicle for constant speed and the power that is consumed during actual driving. At the end, they show, via regression analysis, that these indices can predict variation in fuel economy.

Ping et al., in [35], linked drivers’ behavior to their fuel consumption. They use different parameters such as negative and positive acceleration, along with GPS coordinates and speed. They extract behavior features by applying an unsupervised learning method. Information about the environment is obtained using the onboard cameras and the YoloV3 [36] deep-learning-based object detector. Finally, the environment’s data is combined with driver behavior and fuel consumption data, and is given as input to a long short-term memory (LSTM) neural network, which is another deep-learning-based model [37]. The downside of using a deep-learning-based method is, however, the lack of explainability. Explainability is important for understanding what type of behavior determines a higher level of fuel consumption—and acting accordingly. In our work, we define a very simple feature without any need for a learning-based model, and we prove that it can explain differences in fuel efficiency between drivers.

## 2. Materials and Methods

In order to calculate the AH measurements, we used a dataset of logged information from a fleet of heavy-duty vehicles, including anonymous data about the identity of each vehicle’s drivers. First, it is necessary to extract the geographical locations of the traffic lights in the neighborhoods where the vehicles have been driven. Taking advantage of the vehicle speed and location data, we extract the paths on which the vehicle is approaching traffic lights before stopping at them. Finally, the sequence of brake and acceleration pedal positions on each path is analyzed to extract the AH values. In the following subsections, we describe, in detail, each stage of the AH calculation methodology.

### 2.1. Data

The data used in this study were provided by the Stratio company (https://stratioautomotive.com/, accessed on 13 March 2022) . The logged data contained time-stamped information of several dozens of parameters, including instant fuel consumption, acceleration pedal position, brake pedal position, vehicle speed, and geographical coordinates obtained from the Global Navigation Satellite Systems (https://www.gps.gov/systems/gnss/, accessed on 13 March 2022) (GNSS). GNSS includes multiple satellite constellations consisting of GPS, GLONASS, Galileo, and other geographical positioning systems. In this setting, the approximate accuracy is expected to be within 5 
m
 range, since it is not a high-end system designed for autonomous driving or lane positioning. The log data was collected during three calendar months for a fleet of heavy-duty vehicles. Additionally, anonymized identities for 71 drivers of 40 vehicles were recorded, with time stamps.

The data were gathered using Stratio’s data logger aboard the vehicles. The data logger is connected to their communication area network (CAN) buses. CAN buses are used to establish the communication between sensors and computers onboard the vehicle. Inspecting the CAN signals allows the logger to capture sensor data or even estimated data, such as some actuator positions or brake pad temperature. As vehicles have been enriched with more sensors, the number of CAN buses has increased to meet the data throughput demand. Nowadays, heavy-duty vehicles may have more than eight CAN buses and over 30 onboard micro-controllers.

The ever-increasing data throughput has necessitated the creation of new CAN communication protocols, as well as new proprietary protocols. The latter pose a significant challenge to decoding, requiring decoding parameters that are either given by the original equipment manufacturer (OEM) or obtained through time-consuming reverse engineering.

Stratio’s logger can be simultaneously connected to up to three CAN buses and inspect more than 300 signals. The collected data are then uploaded to a remote server through a 4G data connection. Some signals can be sampled with periods below one second. In order to reduce data transmission costs, the logger compresses the signals onboard using undersampling and a proprietary algorithm.

There are aftermarket and also factory-fit integration schemes to connect Stratio’s DataBox™ logger system to the CANs of the vehicle. One example of the factory-fit integration designed for bus vehicles of the MAN brand is shown in Figure 1a. As can be observed in this case, the logger is connected to diagnostic and body CANs of the vehicle. Additionally, there is a contactless reader which allows for connecting to another internal CAN-bus. There are also connections to ignition switch, power supply, and the K-line, which is a single line serial connection used for diagnostic purposes. Furthermore, a photo of the logger installed on an actual MAN bus is presented in Figure 1b.

### 2.2. Extraction of Red Light Positions

The first step in our proposed method is to extract the positions of traffic lights within the coordinate boundaries of the area used by the fleet. For this task, we take advantage of the overpass-turbo (https://overpass-turbo.eu, accessed on 13 March 2022) tool that can access information from the OpenStreetMap (https://openstreetmap.org, accessed on 13 March 2022) dataset. The traffic light coordinates are used to determine whether the vehicle is, at any given time, approaching a traffic light and stopping. We assume that a vehicle that stopped before a traffic light has encountered either a red light or a visible obstacle. We then extract the data from a fixed traversed distance before the vehicle stops at a traffic light. We denote the fixed distance by 
$L_{p}$
.

### 2.3. Extraction of Red Light-Approaching Trajectories

In order to extract the points that belong to the vehicle’s trajectory before stopping at the traffic light, we first extract the points from the dataset where the vehicle has a slow speed (in our case, less than 1 m/s). The reason for separating these points is that they indicate the occasions when the vehicle has stopped. The speed is calculated from the timestamps and geographical coordinates; however, it could also be obtained directly from CAN data, or from any other reliable source. We keep the coordinates of the traffic lights in a KD-tree for fast access. Using this data structure, we extract the points with slow speed that are closer than 50 m to a traffic light. After that, we iterate through these points chronologically. At each point, we check if there is a path of consecutive points to the closest traffic light. The path could contain points where the vehicle is not moving slowly; however, it must be within a 50 m radius of the closest traffic light from the initial slow point and its last point must be closer than 5 m away from that traffic light. To create the final set of the path points, we start by adding the point closest to the traffic light to the set and iteratively keep adding points that come before that point, one by one, until the length of the path exceeds 150 m. The length of the path is calculated as the sum of the distances between consecutive coordinate points.

More formally, let us assume that 
$C=\langle c_{1},c_{2},\ldots ,c_{N}\rangle$
 is the vector of geographical coordinates of the vehicle during a day. Furthermore, 
$S=\langle s_{1},s_{2},\ldots ,s_{N}\rangle$
 is the vector of vehicle speeds at the corresponding geographical locations, and the order of the points in both vectors is chronological, e.g., the vehicle was at location 
$c_{i}$
 before being at location 
$c_{i+1}$
. Take *T* as the KD-tree that contains the traffic light coordinates. The set of all traffic light paths can be obtained from Algorithm 1. As one can see, the algorithm outputs a set of the paths (*P*) traversed before the stopping points at traffic lights. Each element of the path contains the indices of the data points in the path (*R*) and the traversed distance from those points to the stopping point (*D*). You can see how each path is created in Algorithm 2 which is used as a procedure in Algorithm 1. For each path, the indices in *R* are used to extract the acceleration and brake pedal positions while on the traffic light approach path. Finally, the distances in *D* are used to resample the pedal position values in equidistant intervals, providing 
$N_{p}$
 samples from 0 to 
$L_{p}$
 meters traversed to the stopping point. The re-sampled values for acceleration and brake pedal positions are kept in 
${\langle a_{i}\rangle }_{i=1}^{N_{p}}$
 and 
${\langle b_{i}\rangle }_{i=1}^{N_{p}}$
 sequences, respectively, to be used for AH extraction.
**Algorithm 1 **Finding the set of paths before stopping at red traffic lights during a day for a vehicle**Procedure** FindPaths(C, S, T)    *SP* ← { *i* | *s_i_* ∈ *S* ∧ *s_i_* < 1 }// All point indices with slow vehicle speed    *P*←∅// *P*: The set of extracted traffic light paths    ***For**** i* ∈ *SP*        **If **∄ (*R*, *D*) ∈ *P* such that *i* ∈ *R*            *t* ← *T*(*c_i_*)// *t*: coordinates of the closest traffic light            *j* ← *i* + 1            **While*** j* ≤ *N* ∧ (*dist*(*c_j_*, *t*) < *dist*(*c_j_*_−__1_, *t*) ∧ *dist*(*c_j_,* *t*) < 50)// Distance unit is meters                *j* ← *j* + 1            **End While**            **If** *dist*(*c_j_₋_*1*_*, *t*) < 5//Distance unit is meters                **While** *j* > 1 ∧ (*j*−1 ∈ *SP* ∨ *j* ∉ *SP*)//Go back to the last slowdown point                    *j *← *j* − 1                **End While**                *P* ← *P* ∪ CreatePath(*C*, *j*, *t*, *P*)                **Break**            **End If**        **End If**    **End For**    **Return** *P*
**End Procedure**

**Algorithm 2 **Creating a path given its closest point to the traffic light**Procedure** CreatePath(C, j, t, P)    *k* ← 1    *d_k_* ← 0// *d_k_*: Traversed distance to the stopping point at the path’s *k*-th index    *r_k_* ← *j*// *r_k_*: The data point index at the path’s *k*-th index    **While** *d_k_* < *L_p_* ∧ *r_k_* > 1 ∧ [∄ (*R*, *D*) ∈* P* such that *r_k_*_−__1_ ∈ *R*]        *k* ← *k* + 1        *r_k_* ← *r_k_*_−__1_ − 1        *d_k_* ← *d_k_*_−1_ + *dist*(
$c_{r_{k}},c_{r_{k-1}}$
)    **End While**    *R* ← ⟨*r_*1*_*, *r_*2*_*, ..., *r_k_*⟩    *D* ← ⟨*d_*1*_*, *d_*2*_*, ..., *d_k_*⟩    **If** *d_k_* ≥ *L_p_*// If the path’s traversed distance is greater than *L_p_*        **Return **{(*R*, *D*)}    **Else**        **Return **∅    **End If**
**End Procedure**


### 2.4. Extraction of Attention Horizon

Let us assume that we have a sequence of timestamped data points for a vehicle as it is approaching a traffic light before stopping at it. At each point of the sequence we know the position (measured in percent, where 
0%
 means fully released and 
100%
 means fully pressed) for the accelerator and brake pedals. AH is an estimation of the point at which a driver starts to take action to stop when approaching a red traffic light. It is calculated in meters and can be in the range between 0 and 
$L_{p}$
 meters, i.e., the length of extracted paths.

In general, it is of course difficult to precisely define what kind of indicators should be used to capture a driver’s “decision” in this sense. There is a natural ebb and flow to a vehicle’s speed in traffic, for example, based on the road topology and distances to other traffic participants. In this work, we have chosen to be conservative, and consider lack of acceleration to be an indication of stopping intention. Technically, we define AH as the last point where the acceleration pedal was pressed before starting to slow down, leading to a stop. This can be interpreted formally as

(1)
$$AH=\frac{L_{p}}{N_{p}}l\;,\;l=\left\{\begin{array}{cc} \min_{i\in E}i & \mathrm{if}\;E\neq \emptyset \\ N_{p} & \mathrm{otherwise} \end{array}\right.\;,\;E=\left\{\;i\;|\;\;a_{i}\geq t_{A}\wedge \left[\;\exists \;k\leq i\;s.t.\;b_{k}<t_{B}\;\right]\;\right\}.$$


Here, 
$t_{A}$
 and 
$t_{B}$
 are the minimal thresholds on the acceleration and brake pedal positions, respectively. We assume that the pedals are not pressed when their positions are below these thresholds.

It is also important to note that we exclude from AH calculations those path sequences that have more than one stopping point before the traffic light position. Situations where that happens are usually difficult to interpret, and often indicate traffic conditions where the driver has limited freedom of action. In the end, it leads to significant decrease in the variance of the calculated AH, as explained by different path conditions.

## 3. Results

In order to test the link between the fuel consumption rate and attention horizon, we use heterogeneous naturalistic driving data collected from a fleet of heavy-duty vehicles collected over a timespan of three months. We set the 
$L_{p}$
 value to 150 m and 
$N_{p}$
 to 100; therefore, we resampled and analyzed the brake and acceleration sequences from the 150 m traversed before stopping at the traffic light with 100 equidistant resampled data points.

As previously mentioned, we gathered the anonymized IDs of the vehicles’ drivers throughout the time period. The IDs were employed to associate drivers with the paths with traffic light stops used in AH calculation. We also took advantage of the same IDs to associate drivers with their fuel consumption rate (FCR), i.e., fuel consumed per distance unit. Only 29 drivers had trajectories with stops at traffic lights that could be used to calculate the AH metric.

To give some more insight regarding the calculation of AH, we created plots that show the changes in vehicle speed, and the sequence of positions for the acceleration and brake pedals, through the 150 m just before the vehicle stops at a traffic light. You can see examples of these plots from the same traffic light in Figure 2. Furthermore, we present statistics regarding these parameters at the vehicle approaches to multiple traffic lights in Figure 3. Different behaviors for different approaches to the traffic light can be observed. It is also clear that the AH is essentially the moment the driver stops to accelerate before the last brake.

Our aim in this paper is to show that correlation exists between the AH metric and the overall driver behavior, more specifically one that is associated with long-term fuel consumption. We believe we can use this metric to identify inefficient driving patterns, which would be a very useful service for fleet operators. The operators can then use this information to provide incentives for more efficient and ecological driving and identify training needs, among others. However, to ensure that the influence of other important factors is minimized, we have performed some separate analysis on the AH and FCR values.

To determine the variability of AH based on different factors, we measured its distribution across the traffic lights, and also across the drivers. First, we divided our AH samples based on the driver, and for each driver we calculated their individual average AH at different traffic lights. We created a box plot, capturing the distribution of the data for those drivers who have AH samples originating from more than one traffic light; this way we are showing the distribution of the average calculated AHs across different traffic lights. These box plots can be seen in Figure 4a. After that, we performed a similar analysis for traffic lights by grouping the AHs by traffic light and then calculating the average AH for different drivers that have stopped at each traffic light. We only created box plots for traffic lights with samples from more than one driver. The result is shown in Figure 4b.

The comparison of Figure 4a,b shows a clear difference between the variability of the AH when grouped according to drivers and then averaged for different traffic lights, as opposed to grouped by traffic lights and then averaged for different drivers. It shows a significantly larger variability in the AH distribution across traffic lights than across drivers. Therefore, in the raw data, the influence of local conditions specific to a given traffic light outweighs the influence of different driving styles. This led us to perform a normalization of AHs based on traffic lights, since our goal in this paper is to link AH value to the drivers’ behavior specifically.

To perform this normalization, for each traffic light, we collect all its AH samples and calculate the percentile values for the samples. Then, each AH value is replaced by its corresponding percentile within its traffic light.

We also analyzed the FCR values in a similar manner as for AHs; however, in this case our goal was to compare the effect the driver has on FCR versus the effect the vehicle has. It is obvious that the model, age, and overall condition of the vehicle can have a significant impact on the FCR; in order to provide a fair comparison of the performance of a driver driving it, this should be taken into account. To this end, we wanted to determine how pronounced this effect is for the vehicles in the fleet analyzed in this study.

However, FCR is different from AH in the sense that instantaneous samples of it are essentially useless, since the signal exhibits very high temporal variability and complex (relatively) long-term dependencies. It is only useful if calculated for a time period, typically by dividing the total amount of the consumed fuel by the total distance traversed. It is important to note here that the FCR value can sometimes be misleading. An example could be when the vehicle’s engine is turned on, for maintenance or other purposes, but it is not driven. Hence, to have valid FCR values, we do not include FCR data (distance and fuel consumption) when the speed of the vehicle is below 1 m/s. Another precaution we took was not to take into account FCR data from days when the vehicle traveled less than 10 km at speeds over 1 m/s. The number of vehicles in our dataset that had valid FCR values was 14. The distance limit is intended to exclude short trips. This is because the fleet of the vehicles in our study is entirely made of city and coach buses. These vehicles operate for long hours on a daily basis, and even though they might have rare short trips, those are not representative of their normal usage.

To investigate the effect of the drivers on the FCR, we analyzed this value when the vehicle was driven by each driver. For each driver, we separately calculated the FCR for different vehicles. The distribution of these FCR values for each driver can be seen as a box plot in Figure 4c. On the other hand, the distribution of FCR values for different drivers when driving a specific vehicle can be seen in the form of a box plot in Figure 4d. We should mention that only drivers driving more than one vehicle are represented in Figure 4c and only vehicles driven by more than one driver are represented in Figure 4d.

The comparison of Figure 4c,d shows that, in our data, the variability of FCR values for different drivers across vehicles is higher than the variability of FCR values for different vehicles across drivers. This is the reason to normalize the FCR values by vehicle. The way we normalize FCR depends on the period of time we are calculating the FCR for. For example, if we want to determine the FCR for each day, we calculate the total distance and fuel consumption for each eligible day of each driver–vehicle pair. Then, all these values are grouped based on the vehicle, and their percentile values are calculated for that specific vehicle. Finally, those FCR values are replaced by their percentile index equivalents. If we want to calculate the FCR of a driver regardless of the vehicle, for a period of time (e.g., a day), we obtain the average FCR percentile index for different vehicles for that period of time and driver and then we compute their average, weighted by the distance traveled by each vehicle.

After these normalization steps, we are finally able to investigate the effect of AH on FCR through time, for each driver. In particular, we expect that if a driver has a relatively high (i.e., good) AH during a period of time, it will correspond to a relatively lower FCR. For this reason, for every driver, we took into account normalized FCR and AH values during each day, as described before. Then, we collected these values for all the days and obtained a single correlation value across days between FCR and AH. Since we are calculating correlations, we require drivers with at least two days for which FCR and AH values exist.

We perform the same procedure across weeks and also months to know how long the time period needs to be in order to obtain a good correlation between AH and FCR. We calculated the histograms of these per-driver correlation values and the results are shown in Figure 5. In this figure, we show different histograms when FCR and AH are obtained for calendar days, calendar weeks, or calendar months. The histogram generated for calendar days shows that most of the drivers’ correlations are very close to the value 0. This means that it is not readily possible to connect the normalized AH value of a day to its normalized FCR value for a driver. Too many different factors affect both these measures, apparently. However, as we increase the length of the time period to weeks and months, the histograms show that the correlations progressively shift towards clearly negative values. In fact, the longer the time periods are, the more negative the correlation coefficients become. A negative correlation means that the larger the AH becomes for a person, i.e., the sooner they react to a red traffic light, the lower their overall FCR is.

For extra visualization, we also fitted a normal distribution to the samples of the drivers’ correlation values in each histogram. It can be seen that the standard deviation of the correlation values increases by increasing the length of the time periods. The reason for this could be that more data points become concentrated at the lower end of the possible values’ interval (i.e., −1). Another explanation can be that other factors exist that we have not taken into account. Finally, mean and median of the correlation values are also overlaid on each histogram. The mean and median values for different time periods are additionally shown in Table 1.

It can be seen that both mean and median move closer to −1 the larger the time period becomes. The median of the correlations appears to be decreasing even more than the mean when using months as time period. This, again, can be explained by the fact that the whole distribution of the correlation samples moves to the left for larger time periods; however, since the minimum possible value is −1, the samples become concentrated at the lower end of the possible interval.

In the next section, we discuss our results in more depth.

## 4. Discussion

In the Results section, we showed that the variability of AH caused by different traffic lights is higher than the one caused by different drivers. The difference was so significant that we had to replace the AH values by their corresponding percentile indices within each traffic light. We hypothesize that this high cross-light variability is caused by the different environmental aspects surrounding traffic lights. This could be due to differences in, for example, the visibility of the light on the road, the speed limit before the crossing, the dimensions of the road, its slope, traffic conditions in the area, or any other environmental aspect. Because of this, we believe that normalization based on traffic light is a very important step to make the AH values usable. An interesting path for future work can be investigating the effect of different environmental factors on the AH values.

As we have shown before, the correlation between AH and FCR turns out to be mostly negative for drivers across weeks or months. In other words, a large AH seems to be a predictor of a small FCR within these periods. We suggest that a large AH is a quantitative way of measuring how careful or aware the driver is just before stopping at a traffic light. In this way, AH can be regarded as a metric of general driving efficiency. We found a relationship between AH and FCR, but we would like to propose that AH might also have correlations with other aspects of driving quality. For example, this could include the impact of the driving style on the vehicle’s components wear rate or the frequency of traffic accidents for the driver. This is a very interesting aspect for future research on related driving quality metrics other than FCR.

We have also established that for bigger periods of time (months and weeks rather than days), there are stronger correlations between AH and FCR. We suspect that this happens due to the impact of uncontrolled variables that change in shorter timescales, i.e., noise; as we increase the timescales, the impact of those variables is reduced under the law of large numbers. In this paper, we discovered that we should normalize AH by traffic light and FCR by vehicle. However, there might be other variables that also need to be controlled for in order to improve the AH or FCR calculations. This is another direction of future work for obtaining more precise values for these parameters.

It should also be noted that although for larger time periods the AH-FCR correlation becomes more strongly negative for most of the samples, there are still drivers with positive correlation values. This is an indication that for these drivers, there are still factors whose impact is not smoothed out even when estimating the FCR or AH for timescales of a week or a month. Identifying these factors is another category of future work; in particular, there appear to be some systematic bias factors that are important to understand. This is closely related, but inherently different, from understanding the unknown factors that actually do smooth out during a week or month.

A different avenue for research could be the study of variance in AH value during time. Two drivers could have the same average AH in a period of time but one of them can have much more variance. This could potentially lead to higher fuel consumption for one of them compared to the other one. Considering the variance of AH is another potentially interesting subject to focus on for future work.

Another research direction is investigating the impact of even longer time periods in the AH metric. In this study, we had access to data with anonymized driver IDs for only three months. However, it could be very interesting to know what we could discover if we had data over the period of several years, allowing us to understand how the driving style evolves in the longer time perspective, based on experience.

The scale of the research could be increased in other dimensions, too. One possibility could be, for example, to investigate the impact of a bigger population of drivers and vehicles to see if the correlation values hold up. This might also smooth out even more noise and give a higher quality AH to work with. Another possible research avenue is to compare the extent to which AH explains different driving quality factors across populations, drivers, and vehicles. This implies measuring, for instance, how strong a predictor the AH is when one type of vehicle is used, as opposed to another type. Another option could be to compare the information value of AH across different populations of drivers, e.g., from different areas, occupations, or training background.

Another facet that could be improved about our work is to look for a nonlinear relationship between AH and FCR. Correlation coefficients can only capture the linear component of relationships. Exploring the extraction of nonlinear components between AH and other driver quality metrics is a possible approach for future work.

Finally, it is possible to work on a more sophisticated approach for calculating the AH, e.g., by involving other types of data into its determination. We found our approach to be effective enough but it does not exclude further research in this area. More specifically, in this first work we did not take advantage of machine learning techniques to analyze the data. The definition of attention horizon we used is, intentionally, very simple in order to be interpretable and intuitively understandable. However, using the power of machine learning to mine the available data, one will likely be able to much more precisely pinpoint the elusive point where the driver makes the *decision* to stop at a red light.

## 5. Conclusions

In this paper, we introduced a new metric to analyze drivers’ behavior. We called it attention horizon (AH), since it is inspired by the concept of driver awareness, i.e., how early the driver starts to react when forced to stop at a red traffic light. We presented an algorithm to automatically extract the AH ofgiven positions of red traffic lights in the driving trajectory. The algorithm only needs data points with speed, acceleration and brake pedal positions, and geographical coordinates recorded at regular intervals. Past research on complex notions such as “awareness” typically required highly sensitive data from cameras and detailed human interpretations, while we demonstrate that it is, at least partially, possible to manage at scale with readily-available IoT data.

With simple statistics and without the need for complex tools, we demonstrated that there is a correlation between the AH and fuel consumption rate (FCR) of the drivers in the timescale of weeks and months. This can be used to rank drivers, for example, in terms of their performance.

Furthermore, we demonstrated the necessity of statistical normalization of AH for different traffic lights and normalization of the FCR by vehicle. We presented evidence that the absolute value of the median correlation grows by increasing length of the time periods for which the AH is calculated. In particular, it grows from 0.156 to 0.402 when moving from weekly to monthly periods. We also proposed that the AH could be a general driving quality metric; however, proving that remains to be achieved in future work.

Such a metric as AH that is easy to calculate and straightforward to interpret could be employed as a tool to quantify drivers’ awareness. This can be useful, especially when driving style alone has been shown to account for at least 6% of fuel consumption [38,39], reduced driver vigilance contributes to 35% of motorway crashes [40], and inattentiveness after speeding is the biggest human error in road crashes [27].

In this work, our main focus was to establish AH as a well-founded metric. However, many interesting future research avenues exist based on the proposed AH metric, such as evaluating variance against consistency of the drivers, or analyzing how the AH of an individual driver changes over time, and more. In addition to that, metrics such as the one presented in this paper can be combined with other similar metrics, quantifying driver behavior to create a better driver ranking algorithm. It is important that each of those metrics is as unbiased as possible, and that they have a clear correlation with a desired outcome, e.g., lower fuel consumption, in the case of this paper. Future work can also include developing more of these metrics, capturing other aspects important to driver ranking as well as investigating ways to combine these metrics into a fair driver score.

## Figures and Tables

**Figure 1 sensors-22-02301-f001:**
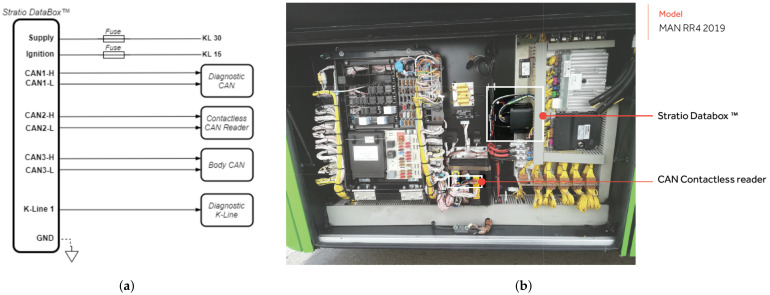
Integration of Stratio’s Databox ™ logger with the CANs of MAN bus vehicles. Shown is (**a**) the connection scheme of the logger with different CAN buses and other connections on the vehicle, and (**b**) a photograph showing the logger installed onboard a vehicle.

**Figure 2 sensors-22-02301-f002:**
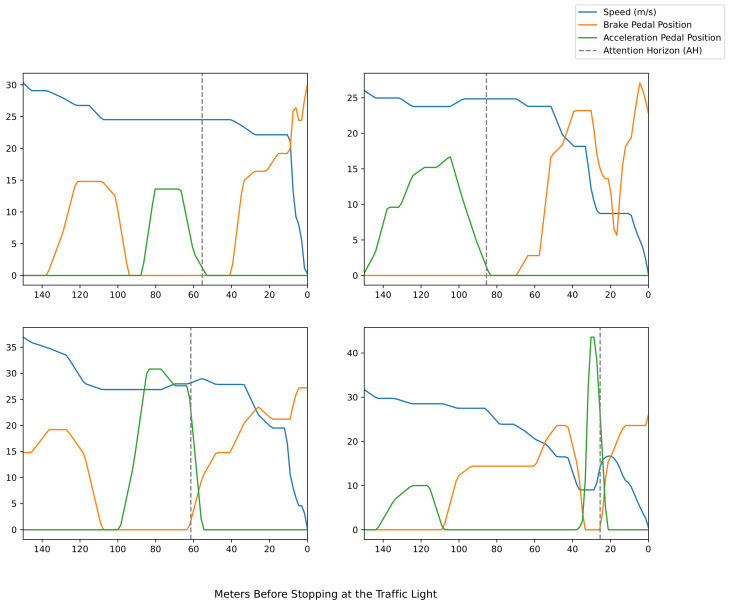
The sequence of acceleration and brake pedal positions, as well as vehicle speeds, just before stops at traffic lights. Four examples from the same traffic light are presented. The calculated AH is shown at the dashed vertical line. Brake and acceleration pedal positions have a possible value range from 0 (fully released) to 100 (fully pressed).

**Figure 3 sensors-22-02301-f003:**
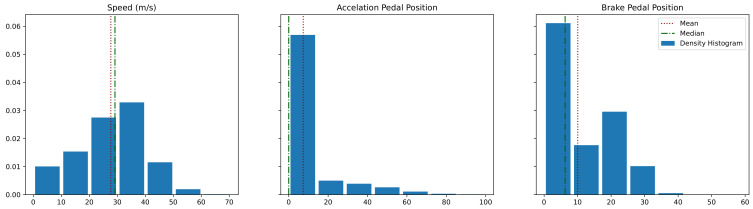
Density histogram, median and mean for the speed, acceleration pedal position, and brake pedal position values for the data point on the traffic light approach sequences (all the data used for determining AH values). Brake and acceleration pedal positions have a possible value range from 0 to 100.

**Figure 4 sensors-22-02301-f004:**
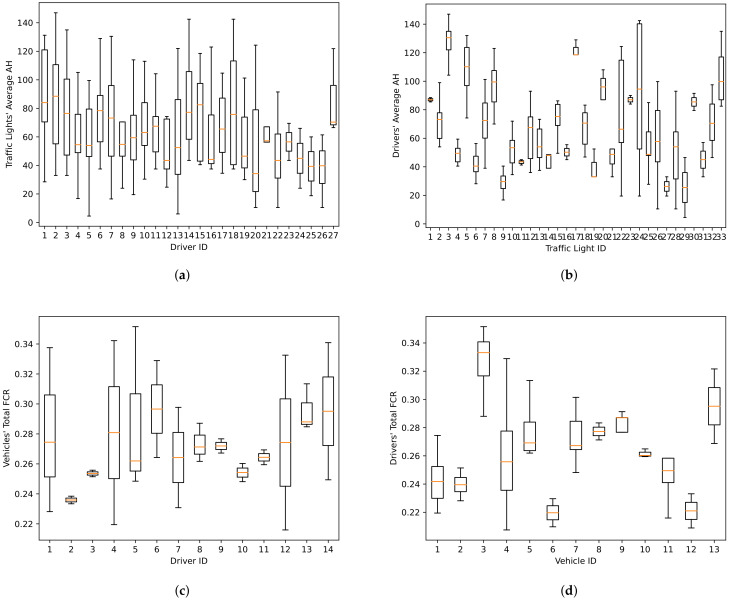
The box plots representing the distributions of (**a**) per traffic light average AHs for each driver, (**b**) per driver average AHs for each traffic light, (**c**) per vehicle FCRs for each driver, and (**d**) per driver FCRs for each vehicle. The box plots for IDs with less than two samples are not shown in this figure.

**Figure 5 sensors-22-02301-f005:**
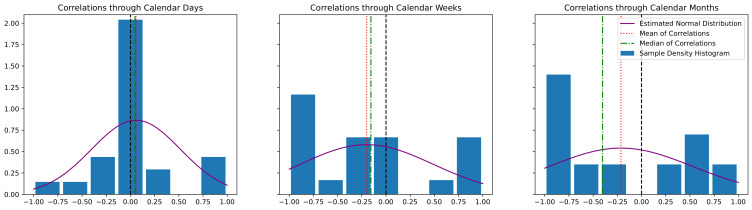
Histogram of correlations between fuel consumption rate (FCR) percentile and attention horizon (AH) percentile through time for individual drivers; these parameters are obtained for days (leftmost), weeks (middle), and months (rightmost plot).

**Table 1 sensors-22-02301-t001:** Mean and median of correlations between AH and FCR through time for different drivers when AH and FCR are calculated for calendar days, calendar weeks, and calendar months.

	Time Periods for AH and FCR Calculation		
	Calendar Days	Calendar Weeks	Calendar Months
Mean Correlation	0.057	−0.200	−0.213
Median Correlation	0.047	−0.156	−0.402

## Data Availability

Not applicable.

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
