# Peer review of "Attention Horizon as a Predictor for the Fuel Consumption Rate of Drivers"

_sensors, 2022, doi:10.3390/s22062301_

Round 1

Reviewer 1 Report

The paper deals with an interesting subject involving the creation of a reaction-based metric and its correlation with fuel consumption rates. The paper uses a simple methodological approach and produces decent results. The manuscript is well written overall. However, there are some conceptual issues in the paper that should be addressed before it is accepted for publication.

  • It would be interesting to mention additional benefits of driver feedback/monitoring, such as driver behavior improvements for road safety. This would increase the impact of the manuscript as it would provide a more compelling argument for driver feedback. Indicative studies are:

Kontaxi, A., Ziakopoulos, A., & Yannis, G. (2021). Investigation of the speeding behavior of motorcyclists through an innovative smartphone application. Traffic injury prevention, 22(6), 460-466.

Soriguera, F., & Miralles, E. (2016). Driver feedback mobile app. Transportation research procedia, 18, 2

  • In Section 2.1, it would be beneficial to have a graphic representation with informative notations on how the CAN bus system is set up to increase comprehension for the unfamiliar readers. Also please mention the accuracy of the system GPS transmissions (e.g. smartphone GPS typically has 5m error, with 15m being the more usual error).

  • In Section 2.2 (line 171), how did the authors verify whether the vehicles stopped due to traffic light only? In other words, how were obstacles filtered out?

  • Οn line 189, how was the 150m threshold determined? Did this number refer to three times the 50m red-light-approach distance?

  • For the process of Section 2.3 and the respective algorithm, did the authors verify the traffic light coordinates? OpenStreetMap data can contain errors. The authors are urged to report how it was ensured that the nearest traffic light was accurately assigned to the vehicle (and not, for instance, a pedestrian traffic light).

  • Please provide descriptive statistics for the quantities shown in Figure 1, as it provides useful insights into the data. For instance, the reviewer was wondering what the min/max/median of brake pedal use percentage were.

  • Οn lines 292-294, the vehicle filtering is mentioned. This process seems too strict as it removes a lot of the urban commuting or routine trips (e.g. business, education, groceries etc.). Furthermore, it leaves the authors with only 14 valid vehicles. Is it possible to revise this limitation, and drop to 2/5 km for minimum trip length, for instance?

  • The authors mention in the abstract that there is high correlation between AH and FCR. However, there is no such evidence reported in the paper. The authors do not provide any metrics to quantify that correlation (such as R^2 values or regression p-values). Please consider providing more metrics, with a univariate regression if possible or similar appropriate analysis.

  • Furthermore, the need to extend the study period into long periods for AH to be correlated with FCR reduces the value of AH as a metric. Aggregated data lose lot of granularity and explanatory power. For instance, a driver with a lot of min-max AH values would be similar to one with average AH values in the long term, however, their driver behavior and FCR would not be comparable. This is a major limitation of the study and should be discussed.

Author Response

We would like to thank the reviewers for their time and valuable comments on our manuscript. You can see the response to your comments in the rest of this letter. Our responses in this letter are written in the blue color for your convenience. Furthermore, the changes in paper are also made in blue to show the parts that are added in response to the reviewers’ comments. 

Before point-by-point responses, we would like to clarify that our study is an observational study performed in a naturalistic setting, hence we were not looking for a cause-and-effect relationship between our metric (AH) and the fuel consumption rate (FCR). It seems that we have created this misunderstanding which we have tried to fix by clarifying the text in our revised submission. 

With that said, we present you with our responses as follows. 

Comments of Reviewer #1: 

The paper deals with an interesting subject involving the creation of a reaction-based metric and its correlation with fuel consumption rates. The paper uses a simple methodological approach and produces decent results. The manuscript is well written overall. However, there are some conceptual issues in the paper that should be addressed before it is accepted for publication. 

1- It would be interesting to mention additional benefits of driver feedback/monitoring, such as driver behavior improvements for road safety. This would increase the impact of the manuscript as it would provide a more compelling argument for driver feedback. Indicative studies are: 

Kontaxi, A., Ziakopoulos, A., & Yannis, G. (2021). Investigation of the speeding behavior of motorcyclists through an innovative smartphone application. Traffic injury prevention, 22(6), 460-466. 

Soriguera, F., & Miralles, E. (2016). Driver feedback mobile app. Transportation research procedia, 18, 2 

Thank you for your suggestion and the proposed indicative studies, we have expanded on the safety aspect of driving in our paper and incorporated your references. 

2- In Section 2.1, it would be beneficial to have a graphic representation with informative notations on how the CAN bus system is set up to increase comprehension for the unfamiliar readers. Also please mention the accuracy of the system GPS transmissions (e.g. smartphone GPS typically has 5m error, with 15m being the more usual error). 

Thanks for your suggestion, we have added an extra figure (now Figure 1) and accompanying text explaining the installation/integration of Stratio’s logger to the CANs of the vehicle. 

We have also clarified information about the geographical positioning system and its error (5 m).  

3- In Section 2.2 (line 171), how did the authors verify whether the vehicles stopped due to traffic light only? In other words, how were obstacles filtered out? 

It is very hard to know whether a vehicle has stopped specifically due to a traffic light, or for some other reason, given the limited data that we have. Many external circumstances affect this and thus (1) we do not know for how long the driver needs to stop at the traffic light and (2) at what distance from the traffic light the stopping occurs. The only heuristic that we could come up with was the distance of the stopping point from the traffic light. We speculated that 50m is a normal upper limit distance from a traffic light at which the vehicle stops. Overall, this is definitely one of the many sources of noise, and the reason why “Big Data” is needed to obtain reliable results – only aggregations over many passes through a given traffic light area will provide reliable information. 

4- Οn line 189, how was the 150m threshold determined? Did this number refer to three times the 50m red-light-approach distance? 

The 150m distance is not related to the 50m distance. It was chosen as the approximate upper limit distance at which we expect the driver to notice, and react, to red traffic light. It is in fact an input parameter to our algorithm (L_p) which could be changed. The main contribution of the paper is to propose the overall approach, which is vastly distinct from methods available in the literature. Many details can, and should, be investigated further in the future by the scientific community. 

5- For the process of Section 2.3 and the respective algorithm, did the authors verify the traffic light coordinates? OpenStreetMap data can contain errors. The authors are urged to report how it was ensured that the nearest traffic light was accurately assigned to the vehicle (and not, for instance, a pedestrian traffic light). 

We use the query ["highway"="traffic_signals"] to extract vehicle traffic lights which is the standard. There is a separate way of encoding pedestrian lights in openstreetmap and that is ["highway"="crossing"] + ["crossing"="traffic_signals"] although it is also possible that they are tagged the same way as the vehicle lights. 

Even in the case that a pedestrian light is tagged as ["highway"="crossing"] there will almost always be another vehicle traffic light very close-by (within a few meters). Hence this does not make much difference for our algorithm, since the vehicle traffic light would be closer to the vehicle stopping point, and it will be chosen as the light. This is because in our algorithm we always take the closest traffic light to the vehicle in our dataset. 

In our algorithm, we also assume that the vehicle has stopped within the 50m distance from the chosen traffic lights which could help filter out wrongly tagged traffic lights. In the end, we are aware that openstreetmap is an open database filled in by volunteers, but it is the best option that we have, and we have taken measures to have accurate positions as explained above. While it would certainly be possible to verify by hand the data we have used in this study, since it is a rather limited geographical region, it would to a large extent defeat the purpose of the study – ultimately, the approach must be fully automatic and highly scalable, capable of compensating for such errors. 

6- Please provide descriptive statistics for the quantities shown in Figure 1, as it provides useful insights into the data. For instance, the reviewer was wondering what the min/max/median of brake pedal use percentage were. 

We regret that we were not clear enough in explaining Figure 1 (now Figure 2). The percentage (%) means the relative position of the pedal, e.g., 0 means that the pedal was not pressed at all and 100 means that the pedal was floored. The percentage value for each of the points in the plot is the position of the pedal at that distance from traffic light only for one instance of a driver stopping at a traffic light. Hence there are no statistics involved. 

We clarified this matter in Figure 1 (now Figure 2) by removing the % sign and instead stating that the possible range is between 0 and 100.  Furthermore, we have added another figure (Figure 3 now), which contains statistics regarding the parameters used to determine the AH value including histogram, median and mean values. 

7 - Οn lines 292-294, the vehicle filtering is mentioned. This process seems too strict as it removes a lot of the urban commuting or routine trips (e.g. business, education, groceries etc.). Furthermore, it leaves the authors with only 14 valid vehicles. Is it possible to revise this limitation, and drop to 2/5 km for minimum trip length, for instance? 

The vehicles in our study are heavy-duty vehicles, in particular buses that operate on a daily basis. Hence, they are not used for small trips (business, education, etc.) while in operation. They might have (rarely) short trips but those are not representative of normal operation, and therefore drivers might operate the buses differently from normal, due to various exceptional circumstances.  

We added a similar explanation to the paper for clarification on this issue. 

8 - The authors mention in the abstract that there is high correlation between AH and FCR. However, there is no such evidence reported in the paper. The authors do not provide any metrics to quantify that correlation (such as R^2 values or regression p-values). Please consider providing more metrics, with a univariate regression if possible or similar appropriate analysis. 

We have provided the histogram for correlation coefficients in our Results section. In our study we claim that there is a negative correlation between AH and FCR we did not claim that we can regress FCR based on AH. We have used Pearson correlation coefficients to show the negative correlations. R^2 and regression p-value is made to quantify regression which is not the claim of our study since correlation could be used for prediction too. 

For more clarification on our results now we have added a table (Table 1) to the paper containing the mean and median of correlation values in addition to the previously existing correlation histogram figure that we had. 

9 - Furthermore, the need to extend the study period into long periods for AH to be correlated with FCR reduces the value of AH as a metric. Aggregated data lose lot of granularity and explanatory power. For instance, a driver with a lot of min-max AH values would be similar to one with average AH values in the long term, however, their driver behavior and FCR would not be comparable. This is a major limitation of the study and should be discussed. 

Thanks for your suggestion. You can be right about the loss of granularity however this is the price we had to pay to remove the noise caused by unknown variables. In the current paper we intend to, primarily, establish AH as a valid metric to consider. There is a lot of interesting future work about analyzing it more in-depth (as you mention, the min-max, high variance drivers against consistent average driver, or changes of AH per driver over time, and more). 

Reviewer 2 Report

Is not a clear major contribution against the state of art. authors should improve the abstract and raise a proper research question

Literature survey is not proper cover - a methodology is needed - Several important and similar papers missing. references are not properly formatted and several have mistakes, e.g like 14. Only 32 is short in a rich publication topic

section 3 needs more details about test conditions, clarify drivers ages, driving experience

at conclusions, authors needs to quantify the savings and quantify the traffic influence and others parameters

I do not see any scientific contribution only a technical work performed

Author Response

We would like to thank the reviewers for their time and valuable comments on our manuscript. You can see the response to your comments in the rest of this letter. Our responses in this letter are written in the blue color for your convenience. Furthermore, the changes in paper are also made in blue to show the parts that are added in response to the reviewers’ comments. 

Before point-by-point responses, we would like to clarify that our study is an observational study performed in a naturalistic setting, hence we were not looking for a cause-and-effect relationship between our metric (AH) and the fuel consumption rate (FCR). It seems that we have created this misunderstanding which we have tried to fix by clarifying the text in our revised submission. 

With that said, we present you with our responses as follows. 

Comments of Reviewer #2: 

1- Is not a clear major contribution against the state of art. authors should improve the abstract and raise a proper research question 

Thank you for the comment. We have clarified the contribution of the paper in the Abstract, Introduction and Conclusions. 

2- Literature survey is not proper cover - a methodology is needed - Several important and similar papers missing. references are not properly formatted and several have mistakes, e.g like 14. Only 32 is short in a rich publication topic 

Thanks for your feedback. To the best of our knowledge, there is no relevant literature to reference from the methodological point of view. This study is the first case where the data necessary to calculate metrics like AH is available at this scale. We will be grateful if you can point out any papers we have missed. Furthermore, we have corrected formatting errors and increased the number of references to 40. 

3- section 3 needs more details about test conditions, clarify drivers ages, driving experience 

We would like to clarify that our study is not a controlled study but an observational study using naturalistic data of actual driving – which we think is a strong point of our work. Hence, we do not have access to driver’s personal information. We were only given access to anonymous identity codes (because of obvious privacy issues, collaborating with a company). However, those are all professional heavy-vehicle (bus) drivers; nevertheless, their experience likely varies a lot. 

4- at conclusions, authors needs to quantify the savings and quantify the traffic influence and others parameters 

More information about the effects of drivers on fuel consumption and traffic safety has been added to the paper. 

5- I do not see any scientific contribution only a technical work performed 

We hope the revised text makes the contribution more clear. We have proposed a new metric that quantifies driver’s “awareness” of their surroundings, suitable for fully automatic calculations based on easy to collect data. Starting from the hypothesis of our metric being related to the long-term fuel consumption, we have shown the result from our observational study that strongly agrees with our hypothesis. 

Round 2

Reviewer 1 Report

The authors have taken into account the previous comments and they have addressed them efficiently and satisfactorily. The paper appears ready for publication and should therefore be accepted.